

# Phenology and predictors of spring emergence for the Timber Rattlesnake (*Crotalus horridus*)

Andrew C. Jesper[1], Scott A. Eckert[2], Brian J. Bielema[3], Scott R. Ballard[4] and Michael J. Dreslik[1]

[1] Illinois Natural History Survey, Prairie Research Institute, Champaign, IL, USA
[2] Department of Biology and Natural Resources, Principia College, Elsah, IL, USA
[3] Unaffiliated, Morrison, IL, USA
[4] Illinois Department of Natural Resources, Carterville, IL, USA

## ABSTRACT

Many temperate reptiles survive winter by using subterranean refugia until external conditions become suitable for activity. Determining when to emerge from refugia relies on the ability to interpret when above-ground environmental conditions are survivable. If temperate reptiles rely on specific environmental cues such as temperature to initiate emergence, we should expect emergence phenologies to be predictable using local climatic data. However, specific predictors of emergence for many temperate reptiles, including the Timber Rattlesnake (*Crotalus horridus*), remain unclear, limiting our understanding of their overwintering phenology and restricting effective conservation and management. Our objectives were to identify environmental cues of spring emergence for *C. horridus* in Illinois to determine the species' emergence phenology, and to examine the applicability of identified cues in predicting emergence phenology across the species' range. We used wildlife cameras and weather station-derived environmental data to observe and predict the daily surface presence of *C. horridus* throughout the late winter and early spring at communal refugia in west-central and northern Illinois. The most parsimonious model for predicting surface presence included the additive effects of maximum daily temperature, accumulated degree days, and latitude. With a notable exception in the southeastern U.S., the model accurately predicted the average emergence day for eight other populations range wide, emphasizing the importance of temperature in influencing the phenological plasticity observed across the species' range. The apparent broad applicability of the model to other populations suggests it can be a valuable tool in predicting spring emergence phenology. Our results provide a foundation for further ecological enquiries and improved management and conservation strategies.

## INTRODUCTION

Seasonally colder periods in temperate regions expose reptiles to temperatures exceeding critical thermal minima (*Gregory, 1982*). A common strategy to survive such critically low temperatures is to retreat into subterranean refugia until external conditions

Corresponding author
Andrew C. Jesper,
ajesper2@illinois.edu

become suitable for activity. Though buffered from external conditions, occupants of refugia are often still subject to thermal regimes that are lower than preferred (*Brown, 1993*; *Claussen et al., 1991*; *Firth & Belan, 1998*), resulting in a cold-induced dormancy (brumation) characterized by restricted physiological, behavioral, and cellular-level functions (*Angilletta Jr, 2009*). Brumation poses several challenges to reptiles. Despite highly depressed metabolic rates, brumation creates an energy deficit, and reptiles must survive solely on stored energy reserves. Brumation also restricts a reptile's ability to conduct vital active-season processes such as foraging, reproduction, or basic physiological maintenance (*Blouin-Demers, Prior & Weatherhead, 2000*; *Smits & Yorke, 1980*; *Viitanen, 1967*). Reptiles must therefore limit the duration of brumation and balance the inherent risks of emerging too early with the advantages of maximizing active duration out of the refugia.

Determining when to emerge from refugia relies on the ability to interpret when above-ground conditions are survivable. Such a decision is particularly complicated in temperate climates because of their stochastic environmental conditions. While studies have proposed several cues for emergence, including physiological thresholds (*Angilletta Jr, 2009*), endogenous rhythms (*Lutterschmidt, LeMaster & Mason, 2006*; *Weatherhead, 1989*), rainfall/humidity (*Viitanen, 1967*), and photoperiod (*Rismiller & Heldmaier, 1982*), the most prevalent for temperate reptiles is temperature. Reptiles generally emerge as air temperatures rise in the spring, a phenomenon correlated with several covariates, including maximum, minimum, and mean daily temperatures (*Bishop & Echternacht, 2004*; *Brown, 1992*; *Graves & Duvall, 1990*), accumulated degree days (ADD; *Hoffman, 2021*; *Turner & Maclean, 2022*) and moving "lagged" average temperatures (*DeGregorio et al., 2017*). Many species emerge only when specific threshold temperatures are surpassed (*Blouin-Demers, Prior & Weatherhead, 2000*; *Burger, 2019*; *DeGregorio et al., 2017*; *Sexton & Marion, 1981*), although significant inter-individual variation has often confounded the identification of a reliable thermal trigger. Such thresholds likely reflect the thermal dependency of many physiological, behavioral, and cellular-level functions, which dictate the lower thermal limits of surface activity (*Angilletta Jr, 2009*).

If spring emergence in temperate reptiles is prompted by specific environmental cues such as temperature, we should expect emergence phenologies to be predictable using local climatic data. Wide-ranging species subject to latitudinal and elevational clines might exhibit phenological plasticity, similar to what is observed for insects (*Cayton et al., 2015*; *Herms, 2004*; *Uelmen Jr et al., 2016*), plants (*Aslam et al., 2017*) and mammals (*Boutin & Lane, 2014*). However, local adaptation or study methods have often confounded the identification of such spatial patterns (*Andrews & Waldron, 2017*; *Blouin-Demers, Prior & Weatherhead, 2000*). While environmental variables, primarily temperature, likely dictate the timing of emergence, specific predictors of emergence for many temperate reptiles remain unclear. Furthermore, no study has examined the applicability of their results across populations, particularly for wide-ranging species exhibiting a high degree of phenological plasticity (*Blouin-Demers, Prior & Weatherhead, 2000*; *DeGregorio et al., 2017*; *Gregory, 1982*; *Martin, 1992*). The lack of phenological schedules for such species

limits our understanding of overwintering ecology and restricts effective conservation and land management (*e.g.*, prescribed burns, tree thinning).

The imperiled Timber Rattlesnake (*Crotalus horridus*) is a wide-ranging pitviper that depends on subterranean refugia for overwinter survival. Dependency on refugia throughout higher latitudes within its distribution dictates the species' ecology— exemplified by communal overwintering of up to 200 individuals (*Brown, 1993*) and seasonal movements between refugia and summer habitat in the fall and spring (*Brown, 1992*; *MacGowan, Currylow & MacNeil, 2017*; *Sealy, 2002*). *Crotalus horridus* has the largest geographic range of any rattlesnake and exhibits considerable phenological plasticity in its overwintering ecology (*Andrews & Waldron, 2017*; *Brown, 1992*; *Martin, 2002*). For example, populations in warmer southern climates emerge up to 3 months earlier (March/April in South Carolina; *Andrews & Waldron, 2017*) than northern climates (May/June in New York; *Brown, 1992*).

Despite studies identifying various temperature-related drivers of emergence (*Andrews & Waldron, 2017*; *Brown, 1992*; *Martin, 1992*), the range-wide applicability of specific cues remains unknown. Consequently, the timing of spring emergence for most populations, such as in Midwest states including Illinois, remains undefined. Such inquiries are particularly relevant for *C. horridus* throughout northern regions where philopatric individuals congregate at refugia during spring emergence and are consequently susceptible to threats including human persecution (*Galligan & Dunson, 1979*) and management activities such as prescribed burns (*Beaupre & Douglas, 2012*). As a slow-maturing species with infrequent reproductive events (*Aldridge & Brown, 1995*; *Bielema, 2022*; *Brown, 1991*), *C. horridus* cannot rapidly recover from population declines. Therefore, knowledge of spring emergence phenology is invaluable for management and conservation purposes.

Our objectives were to: (1) identify environmental cues of spring emergence for *C. horridus* to determine emergence phenology; and (2) examine the applicability of identified cues in predicting the emergence phenology across the species' range. We used wildlife cameras and weather station-derived environmental data to construct a predictive model for the daily surface presence of *C. horridus* during spring egress at two sites in Illinois. We then examined the applicability of our model in correctly determining the spring egress of other *C. horridus* populations by comparing our predictions with the average egress dates reported by other studies. We also used our model to predict surface presence across the latitudinal gradient of Illinois, allowing examination of the intra- and inter-annual differences in predicted (untested) surface presence across a finer latitudinal scale. Our results provide a foundation for further ecological enquiries and effective management and conservation strategies.

## MATERIAL AND METHODS

### Study site and data collection

We conducted research at two over-wintering sites located ~350 km (>3° latitude) apart in western Jo Daviess County (northern Illinois; Fig. 1) and Principia College in Jersey County (west-central Illinois; Fig. 1), Illinois, USA. We performed all research under an
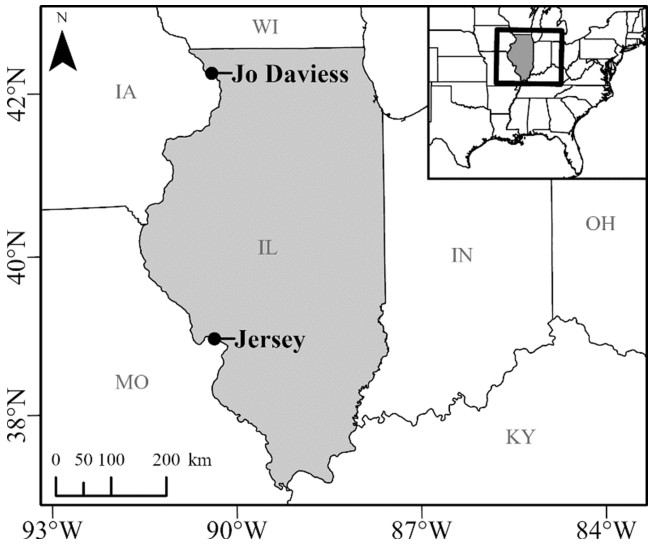

**Figure 1** **The locations of the Timber Rattlesnake (*Crotalus horridus*) refugia we studied in Jo Daviess** (*n* = 3 refugia) and Jersey (*n* = 3 refugia) Counties in northern and west-central Illinois. We monitored the daily surface presence of snakes at all six refugia during the late winter and spring of 2018–2020. Map created in ArcGIS Pro Version 2.7.3 (Esri, Redlands, CA, USA, *2020*).

approved Illinois Department of Natural Resources Endangered and Threatened Species Permit (#05-11S) and approved University of Illinois Institutional Animal Care and Use Committee protocols (IUCUC: 22167 and 22168). Determining refugia locations for the enigmatic *C. horridus* is difficult, and consequently only a few refugia locations are known throughout the state. Such refugia are typically found along larger river valleys and unglaciated regions in northwestern and southern Illinois where outcrops and ridgelines intersect with most of the states' remaining forests (*Bielema, 2022*; *Phillips, Crawford & Kuhns, 2022*; *Smith, 1961*). We chose our two sites because: (1) the latitudinal difference presented an opportunity to examine phenological differences between populations; (2) refugia were situated on private or protected lands which we had permission to access; and (3) we knew of the exact location of refugia entrances (*Bielema, 2022*; S Eckert, 2023, unpublished data).

Knowledge of *C. horridus* at Jo Daviess County dates to the 1930s, with locations of refugia discovered *via* visual encounter surveys in 1991 (*Bielema, 2022*). Known occupancy at Principia College also dates to the 1930s with the acquisition of the land for the College, with specific entrances identified and reconfirmed *via* visual encounter surveys and VHF-radiotelemetry of individuals beginning in 2010 (S Eckert, 2023, unpublished data). Both sites consist of upland mesic forest bounded to the south (Jersey County) and southwest (Jo Daviess County) by Mississippi River limestone-dolomite bluffs, covered with a vegetational matrix of remnant hill prairies and oak-hickory dominated woodlands. Crevices, talus, and holes along the bluff fronts at both sites provide overwintering refugia for *C. horridus*.

We monitored *C. horridus* activity at all identified refugia entrances using Bushnell HD Trophy Cameras (Model #119736) fitted with external 12V batteries for extended life. Our cameras were in remote, topographically rugged locations away from local communities, and thus no people, apart from the researchers, visited the sites or were photographed. Each camera's position depended on substrate and habitat but were generally set ~1–2 m from refugia and at the same elevation as the main entrances to afford a satisfactory field of view. Preliminary investigation revealed cameras occasionally failed to photograph *C. horridus* because the passive infrared sensors (PIR) did not detect slow-moving individuals. Therefore, we supplemented PIR-triggered photos with the camera's time-lapse feature to take date and time-stamped photographs at 5 min intervals for the deployment periods. A 5 min photo interval provided high-resolution monitoring of refugia while also maximizing battery life and camera uptime. One exception to our camera schedule occurred at Jo Daviess County during spring 2018, where cameras photographed the entrances at 1 hr intervals alongside PIR-triggered photos.

## Data analysis

We visually inspected photographs for *C. horridus* and recorded the dates and times of snake sightings. We then converted the snake sightings of each refugium-year combination into a binomial response variable representing the presence of snakes on the surface at refugia (hereafter "surface presence"), where 1 = snake present and 0 = no snake present. We omitted days when cameras malfunctioned or when they were deployed or retrieved. We determined differences between the days of surface presence for each refugium-year combination using bootstrapped means and 95% confidence intervals (CIs). Specifically, we resampled the ordinal days of surface presence for each refugium-year combination with replacement 10,000 times, calculated the mean for each resample, and then determined the mean and 95% CIs based on the resulting bootstrapped resampling distributions (2.5% quantile = lower CI limit; 97.5% quantile = upper CI limit). Non-overlapping CIs indicated informative differences in the effects between refugium-year combinations.

We used Generalized Logistic Mixed Effects Models (GLMMs) in the R package 'lme4' (*Bates et al., 2015*) to examine the effects of environmental variables (Table 1) on the probability of surface presence; that is, the probability of one or more *C. horridus* being present on the surface during a given day. For our study, we limited our analysis to variables derived from weather stations <= 30 km from each site, allowing for relative comparisons across different spatial and temporal extents. Due to the prevalence and apparent importance of temperature as a cue for emergence within the literature, we focused primarily on temperature-derived variables found to be important drivers of surface presence for *C. horridus* and other temperate reptiles, including maximum, minimum, and mean daily temperatures, five-day rolling minimum and maximum daily temperature, day of year, latitude, and accumulated degree days of base 5 °C (ADD). Accumulated degree days is a phenological measure of seasonally increasing cumulative mean daily temperature above a selected threshold and is frequently used to predict phenological events and organismal developmental stages for a variety of taxa (*Boutin & Lane, 2014*; *Cayton et al., 2015*; *Herms, 2004*; *Uelmen Jr et al., 2016*; *Hoffman, 2021*). We calculated

**Table 1 Environmental variables used to construct 32 candidate models predicting the daily surface presence of Timber Rattlesnakes (*Crotalus horridus*) during spring emergence at two sites in Illinois.** We obtained climatic data from two weather stations located < 30 km from the six overwinter refugia in Jo Daviess (northern Illinois; $n = 3$ refugia) and Jersey (west-central Illinois; $n = 3$ refugia) Counties, which we monitored during the late winter and spring of 2018–2020.

| Predictor | Description (unit) |
| --- | --- |
| Min. Temp | Minimum daily temperature (°C) |
| Max. Temp | Maximum daily temperature (°C) |
| Mean. Temp | Mean daily temperature (°C) |
| $Min_5$. Temp | Five-day rolling minimum daily temperature (°C) |
| $Max_5$. Temp | Five-day rolling maximum daily temperature (°C) |
| Accumulated Degree Days (ADD) | Cumulative mean daily temperature above 5 °C (°C) |
| Day of Year | Ordinal date (1 = January 1st) (day) |
| Latitude | Numerical proxy for study site (degrees) |

degree days for each day using the formula: $((T_{max} + T_{min})/2) - T_{base}$, where $T_{max}$ and $T_{min}$ are the maximum and minimum temperature for a given day, and $T_{base}$ is a selected threshold (base) temperature. The selection of the base temperature of 5 °C represented the lowest temperature we observed surface presence of snakes throughout our study. We then summed ("accumulated") the degree day values for each sequential day, starting from 1 January, to calculate ADD over the study period. Before modeling, we Z-transformed (centered and scaled) all variables and tested them for multicollinearity using Variance Inflation Factor (VIF) analysis, removing highly correlated (VIF >= 5) covariates from the same model.

We created a suite of candidate models (Table S1), including a null model (intercept and random effect only) and a fully additive global model, based on *a priori* hypotheses of drivers of surface presence. The dependent variable for each model was daily presence, the random effect structure was refugium nested within the year, and the fixed effects were a combination of the environmental variables. We included latitude as a fixed effect in all models, serving as a numerical proxy for the site, to examine potential differences between the two populations. Our candidate set also included additive and two-way interactive models of the same fixed effect configurations to account for different hypotheses. For example, a significant interaction between latitude and a temperature-related variable might imply that the effect of temperature on surface presence depended on latitude (Jersey County snakes might be present at the surface at different temperatures than Jo Daviess County). In contrast, an additive model might imply both populations responded equally to temperature, although the probability of surface presence between the two sites might differ. Finally, as demonstrated by other studies, we specified all continuous variables as quadratic terms to account for potential curvilinear relationships (*Hoffman, 2021*).

Examination of candidate models using the R package 'AER' (*Kleiber & Zeileis, 2008*) revealed no overdispersion, and therefore we ranked all models using Akaike's Information Criterion adjusted for small sample sizes (AICc) in the R package 'AICcmodavg' (*Mazerolle, 2020*), and then examined the marginal and conditional effects of the most parsimonious model(s) using the R package 'effects' (*Fox & Weisberg, 2018*). We considered parameters

with 95% CIs not broadly overlapping zero as informative predictors of daily surface presence. We back-transformed the top model for interpretation and graphed the predicted values and 95% CIs using the R package 'ggplot2' (*Wickham, 2009*). We examined model fit *via* marginal and conditional coefficients of determination using the R package 'MuMin' (*Barton, 2015*).

We examined the applicability of the top-ranked $AIC_c$ model in correctly determining the spring emergence of other *C. horridus* populations by comparing our predictions with the average emergence dates reported by other studies. Examination of other studies also allowed us to examine the extent of phenological plasticity in spring emergence across the species' geographic range. We limited our comparisons to studies providing a detailed assessment of *C. horridus* emergence (*e.g., Andrews & Waldron, 2017*; *Brown, 1992*; *Martin, 2002*) instead of briefly mentioning general dates with little empirical evidence. For each study, we used our top ranked $AIC_c$ model to calculate predicted probabilities of surface presence for each ordinal day from day 1 to 243, comfortably spanning the entire emergence period at each site, using environmental data gathered from National Oceanic and Atmospheric Association (NOAA) weather stations nearest to the study sites. Because missing data were present in the NOAA datasets, we calculated 15 year "normals" (averages) for estimation using the top model. Doing so afforded complete datasets and determined the typical climatic conditions and probability of surface presence on a given day for each site. Because we were interested in population-level predictions, we held the fixed effect "latitude" at its mean and set the random effects of refugium and year to zero. If our model predictions were accurate, we expected the day of year with the highest (peak) probability of surface presence at each site to correspond to the average emergence day reported by each study. To aid interpretation, we performed a simple linear regression between each study's reported average emergence day and latitude and graphed the results with our model predictions.

Given our model predictions were accurate, we also used the top-ranked $AIC_c$ model to predict surface presence across the latitudinal gradient of Illinois for each year of the study period (2018–2020). Unlike our previous model predictions at sites which were verifiable by the results of other studies, our predictions across Illinois were untested and as such should be treated as hypothetical until verified with empirical data. Nevertheless, the predictions served to examine the potential intra- and inter-annual differences in predicted surface presence across a finer latitudinal scale, while also providing preliminary phenological estimates for potential conservation and management schedules. To determine our site predictions, we derived the same environmental variables as before from weather stations within each degree of latitude in Illinois (37–42°) and used the top model to generate predicted probabilities of surface presence for each latitude-year combination. As before, because we were interested in population-level predictions, we held the fixed effect latitude at its mean and set the random effects of den and year equal to zero. We determined predicted values and 95% confidence intervals using the R package 'lme4' (*Bates et al., 2015*). Using the 'bootMer' function, we refit the model by resampling the dependent variable, daily presence, with replacement 10,000 times, and calculated the predicted values and 95% CIs based on the resulting bootstrapped resampling distributions (2.5%

quantile = lower CI limit; 97.5% quantile = upper CI limit). We present graphs of the daily predictions and 14 day moving averages (for examination of general phenological patterns) against ordinal date for each latitude-year combination and averaged across all three years.

## RESULTS

We deployed cameras at six overwintering refugia (three in Jersey County and three in Jo Daviess County) for one or more years from 2018–2020 (Table 2), accumulating ~473,000 photos throughout the study. In Jersey County, cameras monitored two refugia for three years and a third refugia for two years after being discovered in 2019. All refugia in Jersey County were <1 km apart and situated on tree-covered talus slopes near bluff prairies. In Jo Daviess County, cameras monitored all three refugia for two years in 2018 and 2019, although we removed data from one refugium in 2019 because vegetation restricted the cameras' view and obscured observations. Jo Daviess County refugia were <0.5 km apart and were located on open-canopy outcrops.

The dates and durations of camera deployment varied between refugia and years (Table 2), but all deployments successfully spanned the snake emergence periods at their respective sites. Generally, most camera records showed several weeks of no snake presence on either end of the deployment periods, although some cameras in Jo Daviess County photographed several post-emergent *C. horridus* remaining near refugia entrances at the tail-end of the emergence periods. The individuals typically coiled in crevices within the camera's field of view and were distinguishable from other emerging snakes as they often occupied the same location each day, were the only snakes seen on the photographs at the end of the emergence periods, and in almost all cases, appeared to return to the refugia after several days of no snake sightings (see gap in days of surface presence in Fig. 2). We suspect the snakes were gravid females who frequently remain near refugia after spring emergence until parturition. Because we were interested only in activity related to refugia use, we removed these observations from further analysis (Table 2; Fig. 2).

Examination of bootstrapped 95% means and CIs revealed the daily presence of *C. horridus* at all refugia in Jo Daviess County occurred later in the spring than in Jersey County (Fig. 2). The mean county-level presence for Jo Daviess County occurred on day 136 (16-May) compared to day 103 (13-Apr) for Jersey County. The 95% CIs also indicated intra-county differences in presence days between some, but not all, refugium-year combinations in Jersey County (Fig. 2). Early "one-off" surface presence occurred at all refugia in Jersey County in most years (Fig. 2), with the earliest activity occurring on day 55 (24-Feb). Cameras observed no such early surface presence in Jo Daviess County. Despite early activity, refugia in Jersey County usually exhibited fewer days of surface presence each year (range = 13–27) than in Jo Daviess County (range = 20–41; Table 2).

We used 1,525 camera-deployment days in our analysis to predict the surface presence of *C. horridus* during the late fall and spring from the six refugia (Table 2). Three candidate models received 100% of the $AIC_c$ weights and included additive or two-way interactive effects between ADD, maximum daily temperature, and latitude (Table 3). The most

**Table 2 Refugium by year deployment summaries of 6 field cameras installed at Timber Rattlesnake (*Crotalus horridus*) overwinter refugia during spring emergence at two sites in Illinois, USA.** We deployed Bushnell HD Trophy cameras (Model #119736) at 6 refugia in Jo Daviess (northern Illinois; $n = 3$ refugia) and Jersey (west-central Illinois; $n = 3$ refugia) Counties during late winter and spring of 2018–2020. Columns represent camera deployment locations (County, Refugia); the year of camera deployment (Year); dates of camera deployment (First, Last, Duration); total number of photos taken during deployment (Photos); and the number of days *C. horridus* were photographed (Presence Days) or not (Absence Days).

| County | Refugia | Year | Deployment dates | | | Photos | Presence days | Absence days |
|---|---|---|---|---|---|---|---|---|
| | | | First | Last | Duration | | | |
| | | 2018 | 2/24 | 6/1 | 98 | 27,376 | 27 | 71 |
| Jersey | 1 | 2019 | 1/1 | 6/6 | 157 | 69,260 | 19 | 138 |
| | | 2020 | 1/1 | 5/31 | 152 | 72,002 | 19 | 133 |
| | | 2018 | 2/24 | 5/31 | 97 | 17,914 | 19 | 78 |
| Jersey | 2 | 2019 | 1/1 | 6/7 | 158 | 46,243 | 22 | 136 |
| | | 2020 | 1/1 | 5/31 | 152 | 50,230 | 20 | 130 |
| Jersey | 3 | 2019 | 1/1 | 6/7 | 158 | 36,283 | 13 | 115 |
| | | 2020 | 1/1 | 5/29 | 150 | 67,045 | 24 | 126 |
| Jo Daviess | 4 | 2018 | 4/8 | 6/23 | 77 | 7,041 | 35[*] | 32[*] |
| | | 2019 | 3/17 | 7/13 | 119 | 34,165 | 35[*] | 64[*] |
| Jo Daviess | 5 | 2018 | 4/8 | 6/13 | 67 | 4,567 | 20 | 47 |
| Jo Daviess | 6 | 2018 | 4/8 | 6/13 | 67 | 6,431 | 38 | 29 |
| | | 2019 | 3/17 | 7/10 | 116 | 33,719 | 41[*] | 47[*] |

Notes.
[*]Values represent the number of observed presence days after the removal of days of presumed gravid females (see text).

parsimonious model included the additive effects between model covariates, accounted for 72% of model weights, and was used for all further analyses. The 95% CIs of ADD and maximum daily temperature (quadratic terms) in the top model did not span zero, signifying they had strong explanatory power and were strongly related to the surface presence (Table 4). Conversely, latitude narrowly spanned zero, indicating the parameter had weaker explanatory power.

The additive-only top model, as opposed to the interactive, implied the surface presence of *C. horridus* at both counties occurred at the same values of maximum daily temperature and ADD. However, the probabilities of surface presence were higher in Jo Daviess County than in Jersey County for both variables (Fig. 3). The marginal effects of ADD (holding maximum daily temperature constant; Fig. 3A) revealed an increase in the probability of surface presence to a peak at 277.24 °C, decreasing thereafter, with the high value of ADD reflecting the accumulation of degree days from day of year 1 (1-Jan). The marginal effects of maximum daily temperature (holding ADD constant; Fig. 3), revealed that the probability of surface presence increased with higher temperatures. We did not observe snakes in Jo Daviess County when the maximum daily temperature fell below 11 °C. In Jersey County, snakes remained present on the surface at temperatures as low as 5 °C; however, such occurrences represented only 2.5% (4/155) of all days occurring below
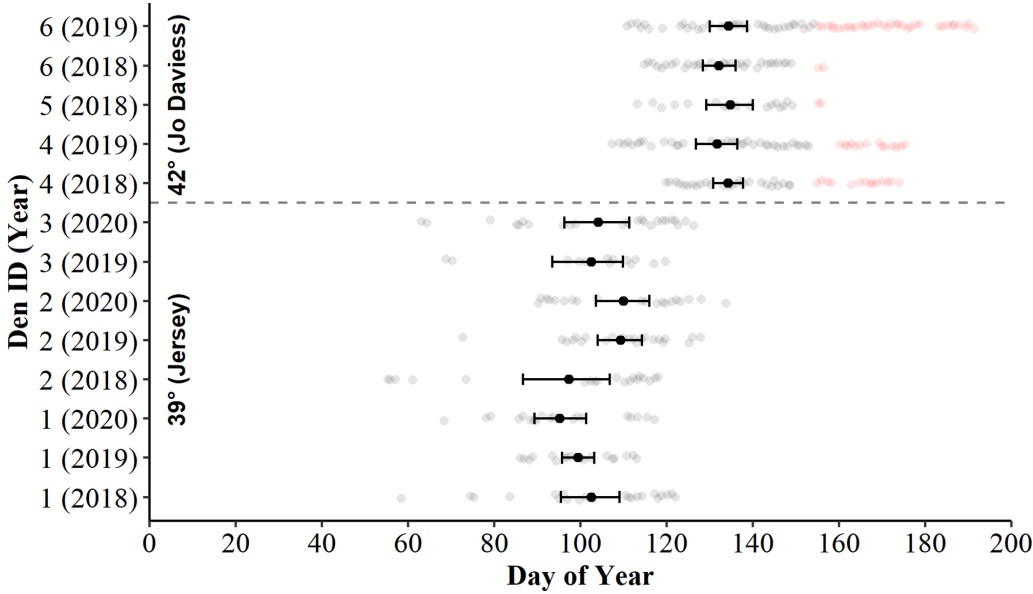

**Figure 2** Days of surface presence for Timber Rattlesnakes (*Crotalus horridus*) during spring emergence at six refugia in Jo Daviess (*n* = 3 refugia) and Jersey County (*n* = 3 refugia), Illinois. We determined days of surface presence (grey and red dots) using Bushnell HD Trophy cameras (Model #119736) deployed at refugia entrances during late winter and spring of 2018–2020. We removed days of suspected gravid females (red dots) from further analysis. The mean day of emergence and bootstrapped 95% confidence intervals (solid black dot and error bars) are displayed for each refugium-year combination.

**Table 3** The top-ranked models examining the effect of environmental variables on the daily surface presence of Timber Rattlesnakes (*Crotalus horridus*) during spring emergence at two sites in Illinois, USA. We used Akaike's Information Criterion corrected for small sample sizes (AIC_c), to rank 32 Generalized Logistic Mixed Effects Models (GLMMs; Table S1) based on a priori hypotheses of drivers of surface presence during spring emergence. The top ten models are shown. We derived surface presence of snakes using field cameras at 6 refugia in Jo Daviess (northern Illinois; *n* = 3 refugia) and Jersey (west-central Illinois; *n* = 3 refugia) Counties during late winter and spring of 2018–2020. Columns represent the number of parameters (k); Akaike score (AIC_c); difference in Akaike score from the top model ($\Delta$AIC_c); Akaike Weights (w_i); loglikelihood (LL); marginal coefficient of determination ($r^2_m$); and conditional coefficient of determination ($r^2_c$).

| Model | $k$ | AIC_c | $\Delta$AIC_c | $w_i$ | LL | $r^2_m$ | $r^2_c$ |
|---|---|---|---|---|---|---|---|
| Latitude + ADD$^2$ + Max. Temp$^2$ | 8 | 788.03 | 0.00 | 0.72 | −385.97 | 0.83 | 0.84 |
| Latitude * Max. Temp$^2$ + ADD$^2$ | 10 | 791.21 | 3.18 | 0.15 | −385.53 | 0.84 | 0.84 |
| Latitude * ADD$^2$ + Max. Temp$^2$ | 10 | 791.47 | 3.44 | 0.13 | −385.66 | 0.84 | 0.84 |
| Latitude + ADD$^2$ + Mean. Temp$^2$ | 8 | 801.19 | 13.16 | 0.00 | −392.55 | 0.83 | 0.83 |
| Latitude * Mean. Temp$^2$ + ADD$^2$ | 10 | 804.12 | 16.10 | 0.00 | −391.99 | 0.83 | 0.84 |
| Latitude * ADD$^2$ + Mean. Temp$^2$ | 10 | 804.73 | 16.70 | 0.00 | −392.29 | 0.83 | 0.83 |
| Global | 16 | 842.46 | 54.43 | 0.00 | −412.17 | 0.81 | 0.82 |
| Latitude + ADD$^2$ + Min. Temp$^2$ | 8 | 850.43 | 62.40 | 0.00 | −417.17 | 0.82 | 0.83 |
| Latitude * Min. Temp$^2$ + ADD$^2$ | 10 | 852.12 | 64.09 | 0.00 | −415.99 | 0.83 | 0.84 |
| Latitude * ADD$^2$ + Min. Temp$^2$ | 10 | 853.22 | 65.19 | 0.00 | −416.54 | 0.82 | 0.83 |

**Table 4 Parameter estimates for the top model examining the effect of environmental variables on the daily surface presence of Timber Rattlesnakes (*Crotalus horridus*) during spring emergence in Illinois.** We used Akaike's Information Criterion corrected for small sample sizes (*AIC$_c$*), to rank 32 (Table S1) Generalized Logistic Mixed Effects Models (GLMMs) based on a priori hypotheses of drivers of surface presence during spring emergence. The top-ranked model included the additive effects of accumulated degree days (ADD), maximum daily temperature (Max. Temp), and latitude. We derived surface presence of snakes using field cameras at six refugia in Jo Daviess (northern Illinois; *n* = 3 refugia) and Jersey (west-central Illinois; *n* = 3 refugia) Counties during late winter and spring of 2018–2020. We obtained climatic data from weather stations located < 30 km from the six overwinter refugia. Columns represent the model parameter (Parameter); the parameter estimate (Estimate); and the standard error (SE); and the upper and lower 95% confidence intervals (CI) of the parameter estimates.

| Parameter | Estimate | SE | Upper CI | Lower CI |
|---|---|---|---|---|
| Intercept | −4.61 | 0.36 | −5.27 | −3.90 |
| ADD | −167.80 | 17.47 | 199.26 | −133.40 |
| ADD$^2$ | −151.52 | 13.99 | −177.20 | −124.71 |
| Latitude | 0.36 | 0.28 | 0.03 | 0.63 |
| Max. Temp | 80.69 | 10.20 | 60.56 | 99.89 |
| Max. Temp$^2$ | −23.02 | 8.24 | −38.37 | −6.53 |

11 °C. The combined effect of both variables implies the probability of surface presence increases with higher ADD and temperatures (Figs. 3C and 3D).

We used seven other studies (eight sites) to examine how accurately the top AIC-ranked model (holding latitude constant; Fig. 4) predicted spring emergence for other *C. horridus* populations (Table 5). Three of the studies determined the timing of spring emergence using daily VHF-radiotelemetry of transmitted individuals (*Andrews & Waldron, 2017*; *Bauder et al., 2011*; *Hoffman, 2021*), three using visual encounter surveys (*Brown, 1992*; *Martin, 2002*; *Sealy, 2002*), and one *via* indicative spikes in body temperature (*Nordberg & Cobb, 2017*). The studies reported the average day of spring emergence spanning latitudes from ∼32.4° to ∼43.8° (Table 5; Fig. 5). Simple linear regression (Fig. 5) revealed a later date of spring emergence as latitude increased ($r^2 = 0.81$). Our model predicted the average emergence day at each site within 10.2 (SD = 13.1) days. Closer inspection revealed our model failed to accurately predict two sites in Hampton (−23 day difference) and Beaufort Counties (−42 day difference), east-central South Carolina (*Andrews & Waldron, 2017*), which inflated prediction error (Fig. 5; Table 5). If we removed the two sites in South Carolina, our model predicted the day of peak emergence for the remaining sites within 4.6 days (SD = 4.2).

We also predicted the probability of surface presence across each latitude-year combination of Illinois (Fig. 6). Visual inspection of each latitude-year subplot suggests the probability of surface presence is highly stochastic throughout late winter and spring, with intra- and inter-year phenological differences within each degree of latitude, particularly in more southerly regions. However, a general unimodal trend is apparent, characterized by a steady increase in the probability of surface presence to a peak as individuals emerged from refugia, followed by a decline in surface presence as snakes dispersed to nearby basking habitat or summer ranges. The peak probability of surface presence between the southernmost (37°; peak probability = day 95) and northernmost latitudes in Illinois (43°;

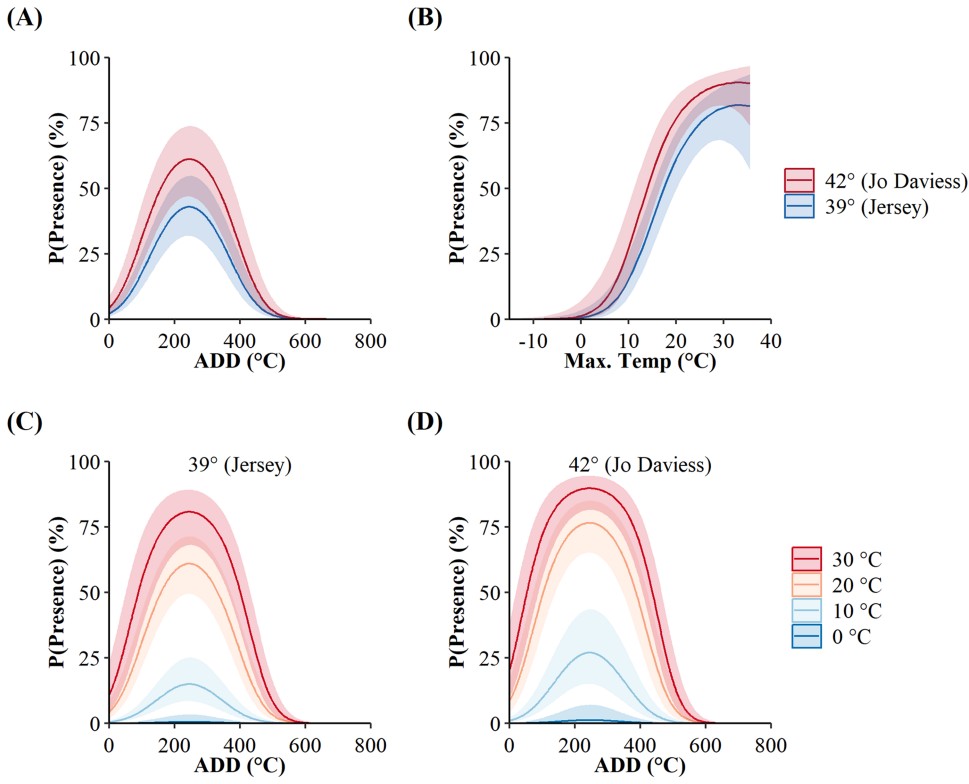

**Figure 3** Site-level probabilities of the daily surface presence of Timber Rattlesnakes (*Crotalus horridus*) at refugia in Jersey (*n* = 3 refugia) and Jo Daviess (*n* = 3 refugia) Counties, Illinois. Predicted probabilities (solid lines) and 95% confidence intervals (ribbons) of the top candidate model as ranked using Akaike's Information Criterion corrected for small sample sizes (AICc). Subplots represent the individual effects of accumulated degree days (ADD; holding maximum daily temperature constant at its mean of 15.95 °C; subplot A), maximum daily temperature (holding ADD constant at its mean of 248.88 °C; subplot B), and the additive effects of both variables for each county, in maximum daily temperature increments of 10 °C (subplots C and D). Data collected during late winter and spring of 2018–2020.

peak probability = day 137), averaged over the three study years, were approximately 42 days apart. Thus, our model suggests a 1° increase in latitude shifted the predicted peak probability of surface presence approximately seven days later into spring, although substantial annual differences in peak surface presence occurred across latitudes, likely due to local climatic variation (Fig. 6). Additionally, comparison between subplots suggests an increased probability of daily surface presence earlier in the season at progressively lower latitudes with early spikes of probability on warmer days, perhaps indicating the greater potential for early "midwinter" emergences (longer left tails and probability spikes; Fig. 6).

## DISCUSSION

We used wildlife cameras and weather station-derived environmental data to successfully observe and predict the daily surface presence of *C. horridus* throughout the late winter

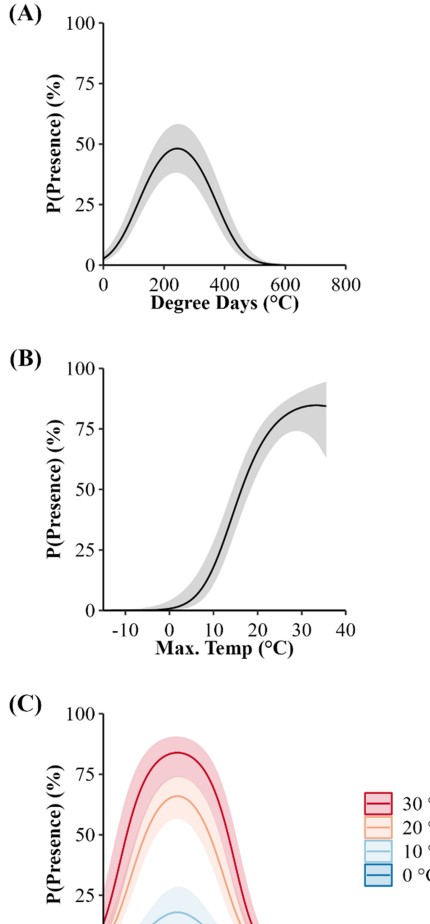

**Figure 4** **Population-level probabilities of the daily surface presence of Timber Rattlesnakes (*Crotalus horridus*) at refugia in Jersey (*n* = 3 refugia) and Jo Daviess (*n* = 3 refugia) Counties, Illinois.** Predicted probabilities (solid lines) and 95% confidence intervals (ribbons) of the top candidate model, as ranked using Akaike's Information Criterion corrected for small sample sizes (AICc), while holding the effect of latitude constant. Subplots represent the individual effects of accumulated degree days (ADD) (holding maximum daily temperature constant at its mean of 15.95 °C; A) and maximum daily temperature (°C) (holding ADD constant at its mean of 248.88 °C; B), and the additive effects of both variables (C). Data collected during late winter and spring of 2018–2020.

and early spring at communal refugia in Jersey and Jo Daviess Counties, Illinois. The most parsimonious model for predicting surface presence included the additive effects of maximum daily temperature, accumulated degree days, and latitude. With a notable exception in the southeastern US, the model accurately predicted the average emergence day for other populations as reported by studies located across the species range, emphasizing the role of temperature in influencing the substantial phenological plasticity observed across the species' range. The apparent applicability of the model to other populations

**Table 5 Summary of the studies we used to examine the applicability of our top-ranked model in correctly determining the spring emergence of other timber rattlesnake (*Crotalus horridus*) populations.** Columns represent the citation ID corresponding to the citation indexes in Fig. 5 (ID), the study citation (Citation); the study location (Study Location), the year(s) in which the study was conducted (Year(s)); the average day of emergence as reported by the study (Reported egress); the day of peak probability of surface presence as predicted by our model (Reported egress); and the difference in days between the reported and predicted values (Diff).

| ID | Citation | Study location (county, state) | Year(s) of study | Determinant of egress | Predicted egress | Reported egress | Diff |
|---|---|---|---|---|---|---|---|
| a | *Brown (1992)* | Warren, NY | 1981–1988 | Visual encounter | 133 | 137 | 4 |
| b | *Bauder et al. (2011)* | Rutland, VA | 2011 | Radiotelemetry | 132 | 145 | 13 |
| c | Current study | Jo Daviess, IL | 2018–2020 | Cameras | 136 | 135 | −1 |
| d | *Hoffman (2021)* | Vinton, OH | 2017–2020 | Radiotelemetry | 114 | 122 | 8 |
| e | *Martin (2002)* | Grant, WV | 1989–2001 | Visual encounter | 135 | 137 | 2 |
| f | Current study | Jersey, IL | 2018–2020 | Cameras | 103 | 109 | 6 |
| g | *Sealy (2002)* | Stokes/Surry, NC | 1990–1997 | Visual encounter | 102 | 103 | 1 |
| h | *Nordberg & Cobb (2017)* | Rutherford, TN | 2011–2013 | Body temperature | 98 | 100 | 2 |
| i | *Andrews & Waldron (2017)* | Hampton, SC | 2002–2004 | Radiotelemetry | 82 | 59 | −23 |
| j | *Andrews & Waldron (2017)* | Beaufort, SC | 2006–2008 | Radiotelemetry | 90 | 48 | −42 |

suggests it can be a valuable tool in generating fine-scaled predictions of the timing of spring emergence for unknown populations across much of the species' geographic range.

Our results suggest temperature-related variables are strong drivers of spring emergence for *C. horridus*. Accumulated degree days allowed our model to capture the general increase in temperature occurring at refugia throughout the late winter and early spring. Accumulated degree days have a long history in phenological predictions of plants (*Boutin & Lane, 2014*), invertebrates (*Cayton et al., 2015*; *Herms, 2004*; *Uelmen Jr et al., 2016*), and to a lesser extent, reptiles (*Hoffman, 2021*; *Turner & Maclean, 2022*). Unlike other time-related variables such as ordinal date and photoperiod (*Martin, 1992*), ADD allows for flexible predictions of daily surface presence by accounting for temperature variation across spatial (latitude and elevation) and temporal (years) extents. For example, degree days will accumulate faster in years and regions exhibiting earlier spring warming. Incorporating such climatic variation in phenological studies is particularly important for species occupying large geographic ranges that are subject to varying thermal regimes and exhibit substantial phenological plasticity.

Including maximum daily temperature with ADD allowed our model to capture the highly stochastic thermal regimes characteristic of temperate climates during gradual spring warming. Other studies have implied daily air temperatures are highly influential and suggest surface activity occurs only once thermal thresholds are surpassed (*Andrews & Waldron, 2017*; *Brown, 1992*; *Martin, 1992*). For example, spring emergence was associated with a maximum air temperature of ∼15 °C in New York, South Carolina, and Virginia (*Andrews & Waldron, 2017*; *Brown, 1992*; *Martin, 1992*). Our results concur with past findings, suggesting 15 °C corresponds to a 50% probability of surface presence, above which surface presence was more likely than not.

It is apparent *C. horridus* only remains present on the surface until ∼11 °C, with some exceptions, perhaps indicating the lower thermal limits of the species and the onset of

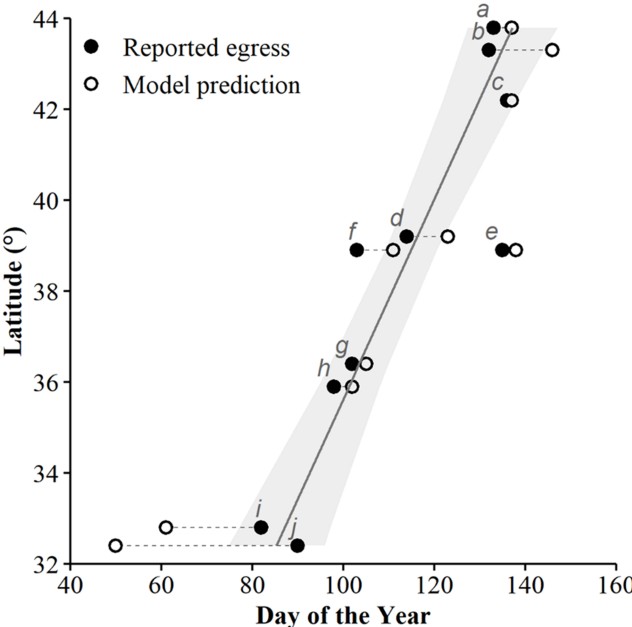

**Figure 5** **The applicability of our top-ranked model in determining the spring emergence of other timber rattlesnake (*Crotalus horridus*) populations.** If our predictions were accurate, we expected the day of year with the highest (peak) predicted probability of surface presence at each site (white dots) to correspond to the average emergence day reported by each study (black dots). Dashed lines connect each study site's reported emergence day and corresponding model prediction. We include a simple linear regression (black line = fitted line; grey ribbon = 95% CIs) between the reported average emergence day and the latitude of each study to examine phenological shifts in emergence timing. Letters correspond to the citation ID of each study detailed in Table 5. We derived predictions using the top AICc model examining surface presence as a function of the additive effects of accumulated degree days (ADD; base 5 °C) and maximum daily temperature (°C) while holding latitude constant. Our predictions were based on data derived from weather station data located within each degree of latitude.

physiological changes which inhibit surface activity. *Agugliaro (2011)* showed temperature-dependent metabolic rate depression in *C. horridus* occurred at 5 °C and 9 °C, with a steep temperature sensitivity in metabolic rate between 9 °C and 13 °C. Similar metabolic sensitivity was found in Red-sided Garter Snakes (*Thamnophis sirtalis parietalis*) between ~10 °C and 15 °C (*Aleksiuk, 1971*). Such metabolic responses likely promote energy conservation during brumation and serve as a mechanism to rapidly return to activity with increasing temperatures (*Agugliaro, 2011*). Snakes tend not to exhibit activity close to their critical thermal limits because of the risks associated with lower performance (*Angilletta Jr, 2009*; *Gregory, 1982*). Thus, the warmer temperature of 15 °C may represent the species' voluntary thermal minima, below which most snakes remain within refugia. However, laboratory-based thermal selection studies on *C. horridus* that emphasize responses to thermal extremes are required to elucidate such thresholds.

Including latitude as a model parameter allowed us to examine the effects of temperature-related variables between sites and across latitudes. Acknowledging that the model included only additive effects, not interactive, between latitude, maximum daily temperature, and

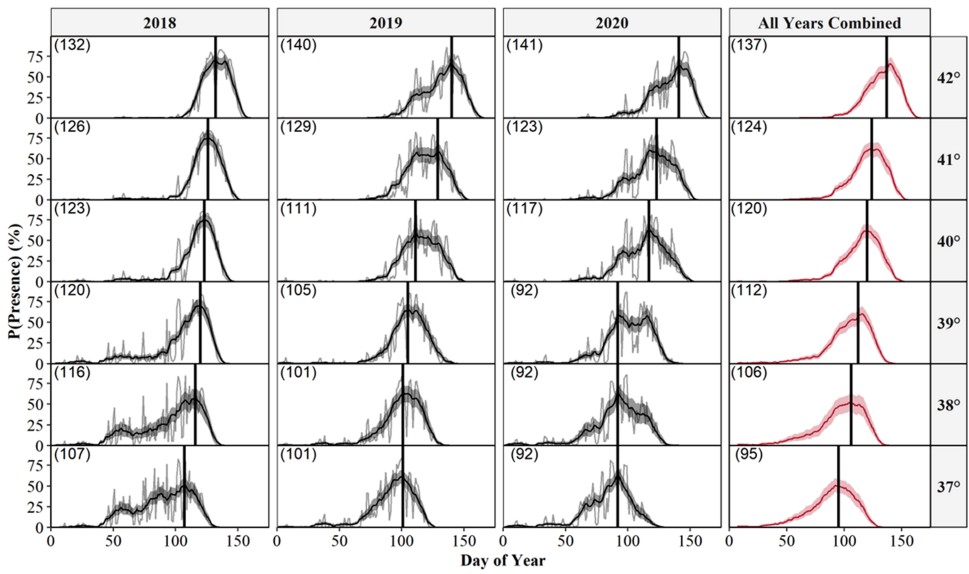

**Figure 6  Predicted daily probabilities of surface presence for the Timber Rattlesnake (*Crotalus horridus*) across the latitudinal gradient of Illinois during the late winter and spring of 2018–2020.** Predicted probabilities are displayed in 1° increments of latitude for each year of our study (2018–2020) and averaged across years. We display probabilities for each day (grey lines) and averaged across a 14 day moving window (black or red lines) with bootstrapped 95% confidence intervals (ribbons). Vertical black lines and parenthesized numbers within each subplot represent the day with the peak probability of surface presence. We derived predictions using the top AICc model examining surface presence as a function of the additive effects of accumulated degree days (ADD; base 5 °C) and maximum daily temperature (°C) while holding latitude constant. Our predictions were based on data derived from weather station data located within each degree of latitude.

degree days is critical for correct interpretation. At both sites, *C. horridus* responded equally to temperature, but the probability of surface presence was higher in Jo Daviess County than in Jersey County. Such a difference likely reflects the distinct habitat configurations at each site, which influenced the post-emergent behavior of *C. horridus* and the cameras' subsequent ability to detect surface activity. Specifically, all refugia in Jo Daviess County were on sun-exposed outcrops with abundant crevices and rock shelves, providing a thermally superior basking habitat and protective cover. Post-emergent *C. horridus* would frequently use such basking habitat, remaining within the camera's view and thus increasing the probability of surface presence.

Conversely, refugia in Jersey County were on closed-canopy, loess-covered talus slopes with a notable lack of undergrowth or rocks near the entrances. Post-emergent *C. horridus* did not linger at the entrances but dispersed from the camera's view to nearby "transient" open habitats such as the bluff front or adjacent hill prairies. Such behavior resulted in fewer daily observations and a lower probability of surface presence. Thus, our model's observed effect of latitude implies the probability of observing snake presence *via* cameras can differ depending on proximate site characteristics rather than intra-population differences in response to temperature.

With notable exceptions in the southeastern US, our top model accurately predicted the average date of emergence for six other populations across the species' geographic range. We should expect some discrepancy between reported *versus* predicted values due to potential sources of error, including the precision of weather station data used for model predictions and the study's sample size and sampling methods providing emergence estimates. Despite these potential sources of error, our model provides accurate predictions for most of the sites examined and indicates temperature is largely responsible for the phenological plasticity of spring emergence exhibited by *C. horridus*. Consequently, there is a strong correlation between spring emergence and latitude, with spring warming occurring progressively later in northern sites (Fig. 5), resulting in delayed *C. horridus* emergence. Delayed emergence has also been associated with higher, colder elevations (*Martin, 2002*). The reported average mean emergence day 135 (15-May) at a high-elevation site at 1,075 m on the Allegheny Plateau, West Virginia, resembles the northern extreme of the species range, approximately 5° latitude north.

Prolonged cold temperatures and subsequent delays in spring emergence at higher latitudes and elevations (*Brown, 1992*; *Martin, 1992*); result in increasingly shorter active seasons with later emergence and earlier ingress, directly impacting life histories. Shorter active seasons reduce the time dedicated to foraging, thus limiting yearly energy acquisition and subsequent adjustment of energy budgets between growth, maintenance, and reproduction (*Brown, 2016*; *Martin, 1993*). Consequently, alongside potentially cooler temperatures during the active season, *C. horridus* in colder climates tend to exhibit slower growth rates, delayed sexual maturity, longer intervals between reproductive events, smaller offspring sizes, and lower reproductive success (*Aldridge & Brown, 1995*; *Brown, 2016*; *Martin, 2002*). Ultimately, prolonged cold temperatures are likely responsible for a reduction in overall fitness, as seen in the apparent vulnerability of northern populations, including those extirpated from Canada and the northeastern US (*Brown, 1992*; *Environment Canada, 2010*).

Our model failed to predict the average emergence day at the two southernmost sites in coastal South Carolina (*Andrews & Waldron, 2017*). We are uncertain about the reasons for the difference but suspect the region's distinct climate might permit an alternate dormancy strategy that our model does not account for. The study by *Andrews & Waldron (2017)* falls between 32–33° latitude and represents the only populations we examined within the humid subtropical climate of the southeastern coastal plain. Relative to the other study locations, winter temperatures throughout the Lower East Coast are warmer, with average maximum daily temperatures of 15–17 °C in the region's coldest months of January and February (*Andrews & Waldron, 2017*). Because of the warmer climate, degree days in Hampton and Beaufort Counties accumulate faster and reach our model's peak predictions earlier in the season than the observed emergence. According to *Andrews & Waldron (2017)*, a maximum daily temperature of ~15 °C corresponded with the 50% probability threshold of daily surface presence, suggesting surface activity of *C. horridus* could occur on most days throughout the winter months, although they did not comment on any midwinter surface activity.
While relatively rare, we observed occasional midwinter surface activity in Jersey county throughout all months except January. We also witnessed the winter emergences of non-target species, including Northern Copperheads (*Agkistrodon contortrix*), North American Racers (*Coluber constrictor*), and Common Garter Snakes (*Thamnophis sirtalis*). Other studies note winter emergences in several other snake species, including *A. contortrix* (*Sanders & Jacob, 1981*), *C. horridus* (*Nordberg & Cobb, 2016*), Prairie rattlesnakes (*C. viridis*; *Jacob & Painter, 1980*), Eastern Hognose Snakes (*Heterodon platirhinos*; *Plummer, 2002*), and Pigmy Rattlesnakes (*Sistrurus miliarius barbouri*; *May et al., 1996*). Notably, *Nordberg & Cobb (2016)* identified 60 winter emergence events from 13 *C. horridus* in Tennessee based on spikes in body temperature. However, despite the number of observations, relatively little is known regarding the frequency and ecological significance of winter surface activity in snakes. Presumably, such activity becomes progressively less frequent in higher latitudes where persistently low temperatures restrict surface activity, a hypothesis that is partially supported by the lack of early winter emergences in Jo Daviess compared to Jersey County, and the lack of observed winter activity at other high-latitude sites (*Brown, 1992*). Furthermore, our predictions of surface activity reveal a progressively higher probability of early surface presence at lower latitudes in Illinois.

While we can only speculate on the significance of such winter surface activity without further study, we suspect such activity is the exception and not the rule. Most early emergences in Jersey County were "one-off" events, typically characterized by a single snake emerging and basking at a refugium entrance on warmer days which permitted surface activity. The snakes perhaps attempted to elevate body temperature to fight disease or infection (*Clark et al., 2011*; *Nordberg & Cobb, 2016*). *Nordberg & Cobb (2016)* observed over 60 emergence events from 12 *C. horridus* that were surgically implanted with radio transmitters only a few days before their ingress into refugia. Thus, it is possible the surgical incision sites did not fully heal before the onset of brumation and necessitated above-ground basking. Additionally, *Clark et al. (2011)* and *Nordberg & Cobb (2016)* noted snakes emerging with skin lesions early in the spring were not uncommon and may have indicated Snake Fungal Disease. We also observed *C. horridus* in Jersey County with severe skin lesions and contusions of unknown origin that may have motivated snakes to emerge and bask. Although the causes behind early emergences are unknown, the surprising frequency of such events warrants further investigation.

## Conservation implications

Spring emergence is a vulnerable period for *C. horridus*, particularly in northerly latitudes where post-emergent and lethargic individuals congregate at communal refugia and are consequently susceptible to local threats, including management activities (*Beaupre & Douglas, 2012*). Reducing the risk of such threats is vital for effective conservation; for example, scheduling prescribed burn regimes to occur when snakes are less likely to be surface present to reduce potential fire-induced mortality. Yet, the enigmatic nature of *C. horridus*, paired with the apparent phenological variation across both latitudinal and elevational clines, makes determining site-specific spring phenologies difficult and consequently limits conservation. Our model's ability to generate the probability of surface

presence for any given day during spring emergence is, therefore, a valuable tool for defining conservation and management schedules. However, predictions about emergence dates at unstudied sites should be treated as hypotheses until verified with empirical data.

One strategy to direct management schedules, once predictions have been verified, is to define specific probability thresholds which can be translated into dates that are useful for management and conservation. For example, management activities could be conducted near refugia until the probability of surface presence exceeds a selected threshold. From a management perspective, the probability of surface presence is synonymous with risk; a higher probability indicates a greater potential for snake surface presence and exposure to management activities. Therefore, the selection of appropriate thresholds depends on the amount of risk deemed acceptable given a specific application. We provide a variety of date thresholds for each latitude in Illinois (Table S2), as determined from general probability trends (*i.e.*, 14 day averages across all years; Fig. 6) to aid in preliminary conservation and land management scheduling.

Ideally, harmful activities would occur only when there is minimal risk of snake presence (*e.g.*, probabilities <5%; Table S2), corresponding to sustained temperatures below the species' suspected thermal limits of 11 °C. However, such thresholds would likely limit management schedules, particularly in warmer southern regions (below 39° latitude) where warmer temperatures increase the probability of surface presence earlier in the season. In such cases, effective cutoffs must balance the risk of snake exposure with time allocated to management activities. We also encourage flexible scheduling whenever possible to account for intra-year and latitudinal climatic differences, although we recognize such scheduling would require the frequent calculation of model predictions based on current temperatures, which are not as readily accessible or practical as a single fixed date threshold.

We also recognize the need for similar statistical models to determine the drivers and phenology of the species' fall ingress. Like spring emergence, fall ingress is a vulnerable period because snakes congregate around refugia prior to brumation (*Brown, 1993*; *Martin, 1992*). While temperature likely has an effect on the timing of ingress (*Brown, 1993*; *Martin, 1992*), we suspect other factors may also play a role (*e.g.*, the timing of arrival to refugia from active-season habitats), which we are currently examining.

An obvious disadvantage of our study methods is our inability to determine the abundance of snakes present on the surface at refugia using only wildlife cameras. Insight into the number of surface-present snakes would afford a more detailed assessment of the spring emergence phenology of *C. horridus* and the implementation of more effective conservation strategies by incorporating population-level risk assessments. Knowledge of snake abundance would also help differentiate between early "one-off" emergence events by a single snake, particularly in more southern regions, and general spring emergence when most snakes emerge and resume active season pursuits. Both differ in associated risk, which we cannot currently distinguish between. Anecdotal observations of the amount of surface activity seen on the cameras (not reported here) suggest the probability surface presence is positively correlated with surface abundance, although such evidence could be misleading as identifying individuals *via* photographs was not possible. Future research should focus on determining snake abundance to examine population-level risk, but we

acknowledge obtaining such information for this enigmatic species would be time- and energy-intensive, as shown by *Brown (1993)* and *Martin (1993)* who spent upward of a decade obtaining such data.

## ACKNOWLEDGEMENTS

We express gratitude to faculty and staff of the Illinois Natural History Survey's Population and Community Ecology (PACE) lab, and the Biology and Natural Resources (BNR) Department at Principia College, for providing support and resources throughout the study. A special thanks to I Armesy and S Myers for their assistance with fieldwork. Finally, thanks to C Suski, M Ward, and J Crawford, alongside the editors and reviewers of PeerJ, for providing valuable feedback on the final manuscript.

### Funding

This work was supported by the Illinois Department of Natural Resources, the Illinois Tollway, and the Illinois Natural History Survey. The funders had no role in study design, data collection and analysis, decision to publish, or preparation of the manuscript.

### Grant Disclosures

The following grant information was disclosed by the authors:
The Illinois Department of Natural Resources, the Illinois Tollway, and the Illinois Natural History Survey.

### Competing Interests

The authors declare there are no competing interests.

### Author Contributions

- Andrew C. Jesper conceived and designed the experiments, performed the experiments, analyzed the data, prepared figures and/or tables, authored or reviewed drafts of the article, and approved the final draft.
- Scott A. Eckert conceived and designed the experiments, authored or reviewed drafts of the article, and approved the final draft.
- Brian J. Bielema performed the experiments, authored or reviewed drafts of the article, and approved the final draft.
- Scott R. Ballard performed the experiments, authored or reviewed drafts of the article, and approved the final draft.
- Michael J. Dreslik conceived and designed the experiments, authored or reviewed drafts of the article, and approved the final draft.

### Animal Ethics

The following information was supplied relating to ethical approvals (i.e., approving body and any reference numbers):

We conducted all research under approved University of Illinois Animal Care and Use Committee protocols (IUCUC: 22167 and 22168).

## Field Study Permissions

The following information was supplied relating to field study approvals (i.e., approving body and any reference numbers):

Fieldwork was conducted on the property of Principia College, Elsah, in Jersey County, Il, and land managed by the Illinois Department of Natural Resources in Jo Daviess County, Il. Research methods and collection were approved by both parties.

## Data Availability

The data is available at figshare: Jesper, Andrew; Bielema, Brian; Dreslik, Michael J.; Ballard, Scott; Eckert, Scott (2023). Data Files for the paper entitled ''Phenology and predictors of spring emergence for the Timber Rattlesnake (*Crotalus horridus*)''. figshare. Dataset. https://doi.org/10.6084/m9.figshare.22653772.v2.

## Supplemental Information

Supplemental information for this article can be found online at http://dx.doi.org/10.7717/peerj.16044#supplemental-information.

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
