# Peer review of "Phenology and predictors of spring emergence for the Timber Rattlesnake (Crotalus horridus)"

_PeerJ, doi:10.7717/peerj.16044_

## Round 0.1 · original submission · Minor Revisions

Dear authors,

We are pleased to inform you that both reviewers have acknowledged the high quality of your manuscript, praising its well-crafted writing style and intriguing analyses. However, they have provided valuable suggestions for minor revisions in the introduction and materials and methods sections. Additionally, they have made some insightful observations that will further enhance the manuscript's chances of acceptance.

We would like to take this opportunity to congratulate you on the exceptional work you have done.

Best regards,

Armando Sunny

Reviewer 1 ·

Basic reporting

This paper describes a detailed and comprehensive modeling approach to understanding the specific climatological factors that drive spring emergence from brumation in the timber rattlesnakes, a wide-ranging reptile of significant conservation concern in the eastern US. The paper is very well written, with a concise but detailed and logically structured narrative describing the background, justifying the methodological approach, and putting the results in context.

Experimental design

The authors use a sophisticated and appropriate modeling framework, and their results represent an important move forward in a developing a more precise understanding of this critical life history event.

Validity of the findings

It’s clear this paper is a useful and important contribution to the conservation ecology literature, and the authors should be commended. I have only a few relatively minor suggestions for improvement.

Additional comments

One question I had regarded the relative population sizes at the different sites in Illinois. Given that the den sites are well-studied, is there evidence that they all support a roughly similar number of individuals? It strikes me that the presence or absence of snakes on the surface is somewhat dependent on population size of the local hibernacula, as large populations would be more likely to have early emerging outliers and stragglers. Widely divergent population sizes between the sites might affect the comparisons.

I would also suggest that the authors at least briefly discuss the potential use of their modeling approach for also studying ingress. Like egress, it is likely driven by local temperature effects, and it would be beneficial for management strategies to be able to predict when ingress has been completed.

More minor comments:

Line 44: cold-induced dormancy is somewhat clunky when used repeatedly throughout the ms—why not use the term “brumation”?

Line 82: when citing cutoffs days for management, perhaps a more specific example could be used for a management intervention that you don’t want to do too close to egress (i.e., tree thinning, controlled burns, etc.)?

Line 85: awkward phrasing, would suggest “…exhibiting population declines” be reworded and expanded to its own sentence or independent clause

Line 103: suggest “…making mortality events…”

Line 208: missing words at end of sentence, I think you mean “…increasing maximum daily temperature.”

·

Basic reporting

I enjoyed reading this manuscript. The authors have done a good job describing why this study is important, how they designed and executed the study, and they have for the most part clearly communicated their results (minus some minor areas noted below). The paper will make a nice contribution to timber rattlesnake literature and it has the potential to directly benefit this species’ conservation. I believe that most of my concerns can be addressed with relatively simple revisions of certain portions of the paper. The writing is generally good, but I have highlighted particular things to watch out for in the detailed comments below (also in the attached PDF with handwritten comments). Adopting a more active voice, avoiding the unnecessary use of nouns as adjectives, and using geographic descriptors that are more meaningful to international readers will improve the final product.

Experimental design

This study does a great job of quantifying the phenology of spring emergence at 2 sites (6 hibernacula) in Illinois and comparing models to identify the most parsimonious combination of variables describing the timing of emergence. The obvious next question is always: How good is the model at predicting behavior outside of the original study sites? The fact that the authors took a stab at this by comparing predicted values from the model to published emergence dates from a few other parts of the range adds quite a bit of value to the paper, and I commend the authors for doing so.

The authors also used their top model to create detailed predictions for the latitudes found within Illinois. I found this last portion of the analysis to be somewhat less useful than the preceding parts, in that we are left with no way of knowing how accurate the predictions were (outside of the general assessment that drew from a few sites outside of Illinois). However, this is not a fatal flaw. The fine-scaled predictions are interesting because they help to show how latitude and annual temperature changes influence predicted emergence dates. The authors just need to be more explicit about why they conducted the fine-scaled predictions (as it is currently presented there is room for confusion), and they should be careful to not read too much into the fine-scaled predictions.

Validity of the findings

Although the relationship between temperature and emergence by the study species is generally understood, this paper nicely demonstrates that the relationship (combined with latitude) is remarkably consistent across most of the populations that have so far been described in the literature. I have no concerns with the major findings in this paper. However, I do think the paper would benefit from some more careful wording when discussing the accuracy of the model predictions. The model clearly has some applicability outside of Illinois, but I would caution the authors that there was a small number of published sites to compare to, and as they noted sites in South Carolina did not conform to the predicted dates of emergence. There’s good reason to expect the same would be true for other sites in the South (and possibly elsewhere). We simply don’t have data from many sites. This may be especially relevant because the southeastern Coastal Plain states of the US probably comprises a large portion of the range-wide population. Thus, I don’t know how I feel about the idea that wildlife managers should use this model to determine when they should or should not conduct risky management activities. This is a preliminary model and there’s probably a lot of geographic variation we’re blind to here. I think it will be up to follow-up studies to “ground truth” the model.

Additionally, initially I was confused why the authors bothered to predict emergence phenology across Illinois, given they did not attempt to test these fine-scaled predictions. I also found myself wondering how many other over wintering sites there are in the state. I’m guessing that data about emergence phenology simply is not available at other Illinois hibernacula. I would suggest revising the introduction and methods to clarify what the authors hoped to gain from this portion of the analysis. After reading the paper a second time I believe that the main value of the predictions is that they demonstrate the interplay between latitude and inter-annual variation in temperature (as opposed to predicting the timing of emergence so wildlife managers can know when it is safe to conduct prescribed burns). Some clarification here will go a long way toward improving the paper.

Additional comments

General Comments
I commend the authors on some nice work. Please see the attached scanned PDF for some hand-written comments directly on the manuscript, in addition to the typed summary below.
Major suggestions (not in order of importance):
1. In the questionnaire that addresses permits to conduct field work, I was expecting to see a copy of the permit or some other documentation. Instead there is a some text that appears to be unrelated to the question.
2. Consider providing a small map that shows the approximate location (county level) of the study sites that were used to assess model accuracy. You might be able to include it as part of Fig. 5. Alternatively, you could use such a map to replace Fig. 1. It would be helpful for readers to be able to see all of the approximate locations, otherwise readers will have to look up the information by reading the original studies. Along the same lines, please specify where these studies took place the first time you mention them in the introduction or methods. I think this would be especially helpful for readers who do not live in the US. We should strive to make our papers approachable to as large of an audience as possible.
3. Similarly, there are a few places in the paper where the wording could be changed to make the paper more approachable to readers from outside the US. Try to not have a US-centric viewpoint. This species occurs in Canada too, right? Consider using North America instead of the US as your frame of reference. Also, it’s okay to state county names in the methods but most people outside of the states in question will have little idea of what you’re talking about, so better to avoid it after the methods. Instead, use the name of the state and a more meaningful descriptor like “northern or southern” or “coastal or mountainous.” I’m thinking specifically here about the Illinois counties as well as those from South Carolina that are mentioned in the discussion. I would extend this line of thinking throughout the paper. Example, for Fig 3 replace the label Jo Davies with “northern Illinois”, and Jersey with “southern Illinois”
4. You use “egress” and “emergence” interchangeably in this paper. For the sake of clarity choose one and be consistent throughout. Emergence seems like a more widely used term, across vertebrate taxa.
5. Please go back and look at each figure and table legend to make sure they all have enough information to stand alone. Example, Fig. 1 should provide inclusive dates. The legend for Fig 5 is missing a lot of information.
6. Some of the tables could probably be included as supplemental information, rather than a standard table.

Minor/detailed suggestions (comments are preceded by the line number):
Ln 16 – replace “overwintering in” with “using” to avoid having the word winter appear twice in the same sentence.
Ln 18 – change to read “relies on the ability to”
Ln 20 – delete the word “available” – this goes without saying.
Ln 23 – I think this should read phenology not “phenologies”
Ln 24 – Objectives were to identify environmental cues that the study species uses to time emergence, not potential cues.
Ln 25 – Technically you did this for snakes in Illinois and then examined whether these same cues were useful at other locations
Ln 28 and 29 – Change this to Northern and Southern Illinois for the sake of readers who do not live in Illinois.
Ln 32 – insert “several” (or perhaps the actual number) before “other populations” so this is not taken to mean you did a truly range-wide analysis. A range-wide analysis implies a more thorough analysis than was possible with the number of studies you had to draw from.
Ln 32-33 – Change to “the importance of temperature in the phenological plasiticity”
Ln 36 – Change “effective” to “improved”
Ln 42 – change to “thermal regimens that are lower than preferred”
Lin 49-52 – Change to read “Reptiles must therefore limit the duration of cold-induced dormancy while balancing the risks…” Delete the last portion of the sentence – it seem unnecessary. These changes are intended to make the sentence shorter and read in a more active voice.
Ln 53 – Change to “relies on the ability to”. Reptiles is unnecessary and this avoids using an apostrophe. Also, the sentence is applicable to more than just reptiles.
Ln 55 – Change to ‘climates because of their stochastic’
Ln 70 – Delete ‘available’ – it is not needed
Ln 77 – I hate when authors say “to our knowledge.” It’s the job of the author to have done a thorough enough literature search to know whether this is the case or not. Saying “to our knowledge” usually only serves to undermine confidence in the paper.
Ln 82-83 – Delete “particularly for species of conservation concern”
Ln 84 – change to “terrestrial pitviper that depends on” – it may have slightly more words but it is easier to read.
Ln 85 – change to “overwinter survival, and has exhibited population declines.”
Ln 89-90 – delete first part of sentence and revise to begin with “Crotalus horridus has the largest geographic range of any rattlesnake and exhibits considerable…”
Ln 91-92 – Change to read “Populations in warmer southern climates emerge..”
Ln 102-104 – The word plasticity shows up a lot in this paper. In this case, “demographic plasticity” is a little ambiguous. A term such as “reproductive rate” seems more appropriate. Change to read “lacks the reproductive rate needed to rapidly recover from population declines..”
Ln 103 – The last part of the sentence that starts with “leaving mortality..” seems unnecessary and it can be deleted without impacting the meaning of the sentence. If they lack the ability to rapidly recover from population declines the fact that it is detrimental to their population viability goes without saying.
Ln 106 – Delete “Thus” and simply state “Our objectives were to..” Also, it seems like you had 3 objectives: 1) to quantify when the species emerges at your study sites in Illinois, 2) to identify variables that best predict emergence, and 3) assess whether the most parsimonious model accurately predicts emergence using published data from elsewhere in the range.
Ln 108-110 – Is this accurate? This sentence implies that you collected data from across Illinois. I think you should be more careful here. You used the model to make predictions about emergence across Illinois, but you only collected data from two sites.
Ln 110-112 – This sentence also confused me a little. I think its worth clarifying somehow that you predicted emergence for other parts of the range and within Illinois, but you only tested how accurate this model was at those sites in other states where there were published data allowing this. The point being you’ll need to make sure the reader doesn’t expect to see model validation based on other populations in Illinois.
Ln 116-129 – Are there other documented refugia in Illinois? How many and where? A little more information here would be good to help us contextualize. If there are other sites, how and why were your study sites selected? I assume because they provided the largest possible latitudinal range, but this needs to be stated.
Ln 133 – delete “involved in data collection (the authors)”
Ln 135 – delete “the” before refugia. Also, because refugia is plural “main entrances” seems more appropriate than the singular.
Ln 136-137 – I would rephrase to the following: “..failed to photograph C. horridus because the passive infrared sensors did not detect this slow moving species.”
Ln 139 – Change to read “date and time-stamped” and end the sentence after “deployment period.” Begin the next sentence with “A 5 min photo interval provided…”
Ln 144 – delete “all” before photographs (goes without saying)
Ln 145 – Should this be “refugium-year”? Why make refugia plural if you are not doing so to year? This term implies a particular refugium during a particular year, so it seems like they both should be singular. Whatever version you use, please make sure you are being consistent throughout the paper.
Ln 145 - Change to read “surface presence of snakes for each refugia-year combination..” Note: Be careful with the use of terms such as “surface presence” and “surface active.” Using nouns as adjectives should be avoided to the extent possible. “Surface presence” implies that there is a surface present..if interpreted literally. The reader is less likely to get confused if you say “presence of snakes at the surface” and “active at the surface”. Excessive use of nouns as adjectives creates jargon and makes the paper less understandable. I think it’s okay if you retain the use of “surface presence” throughout the paper but I would definitely avoid the term “surface active”. See line 157 for example.
Ln 147 – this should read “days when cameras malfunctioned” not “where”
Ln 147-148 – to avoid the use of slash marks I would revise the last part of this sentence to read “or when they were deployed or retrieved.”
Ln 157 – Change to read “probability of surface presence; that is, the probability of one or more C. horridus being active at the surface on a given day.”
Ln 162 – Change to read “reptiles, including:..”
Ln 164 and elsewhere – Most journals do not want you to start a sentence with an abbreviation. Spell out ADD.
Ln 169 – I think the symbol T should be italicized here, to be consistent with the formula shown on line 168.
Ln 171 – change to read “surface activity of snakes”
Ln 177 – insert a comma between “model” and “based on”
Ln 178 – delete “the” before “daily presence”
Ln 179 – change to “refugium nested within year”
Ln 182-183 – separate out the parenthetical content to make a new sentence.
Ln 185 – change to “..imply that the effect of temperature..”
Ln 186 – active at the surface at different temperatures
Ln 196 – Change to “We considered parameters with…not broadly overlapping zero as informative predictors..”
Ln 204-206 – Where were these studies conducted? Provide us with a little more information here so the reader doesn’t have to look up this information outside of your paper. Also, what methods did these studies use? This is important and potentially relevant to the interpretation of some of you results.
Ln 211 – Data are plural, change to read “data were present”
Ln 212 – Delete the word “parameters”
Ln 220-229 I think that this needs a little explanation or clarification in the introduction. Making these predictions for the sake of understanding the interplay between yearly temperatures and latitude seems like a valid approach, but I’m a little skeptical of predicting surface presence for other reasons, given that you only did some preliminary model validation using a few study sites outside Illinois (i.e. there is no way to validate your predictions within Illinois).
Ln 241 change to “although we omitted data from one refugium”
Ln 251-253 Please explain what criteria you used when deciding to remove these observations? At what point did you consider the snakes as being residents as opposed to those emerging late? Did you use an arbitrary cut-off, was it based on the literature, or some visual evidence? We need a little detail here.
Ln 256 – Delete “day” so this reads “mean county-level presence”. Day is redundant given the rest of the sentence.
Ln 269 – Is this referring to both forms of maximum daily temperature? Please clarify.
Ln 273-274 – I’m not sure that you can say they “responded equally” given how different the parameter estimates were. They responded to both.
Ln 277 – two decimal places for a temperature value of 277 seems like overkill. One decimal place is probably sufficient.
Ln 280 – There is a word missing in the sentence.
Ln 282 – change to “at temperatures as low as 5..”
Ln 286 – change to “to examine how accurately the top AIC-ranked model predicted spring emergence...”
Ln 293 – change to “the two sites in South Carolina, our model predicted the day of peak emergence for remaining sites to within..”
Ln 300 – remove the apostrophe from individuals, and change the rest of the sentence to read “emergence from refugia, followed by a decline in surface presence as snakes disperse to summer habitats.”
Ln 302 – change to “northernmost latitudes in Illinois”
Ln 303 – insert a comma after “three study years”
Ln 304 - change “shifts” to “shifted”
Ln 305 – delete “the” from before “spring”
Ln 306 – change to “likely due to local climatic”
Ln 312 to 314 – This simply repeats what has already been stated elsewhere in the paper. It reads as a methods statement. Reword. I think the more important point you are trying to make is that your work demonstrates that wildlife cameras are useful for monitoring emergence of C. horridus.
Ln 317 – “the model accurately predicted the average egress day for other populations range wide” needs to be reworded for clarity. You only assessed a few populations outside of Illinois – I would not describe this as a “range wide” assessment. That implies a much more comprehensive dataset.
Ln 319-322 – the last portion of this sentence “as demonstrated across Illinois” communicates something that is incorrect. Yes, your top model did a decent job at predicting emergence dates for most of the other populations described in the literature, but the predictions for the various latitudes in Illinois were not tested against observed data and therefore they tell us nothing about the model’s validity. Thus, all the Illinois predictions show is simply that you can use the model to generate fine-scaled predictions. If that is what you meant, please reword accordingly.
Ln 323-326 – spell out ADD when beginning sentences with it.
Ln 325 – the word “sites” is redundant, you can simply say “refugia”
Ln 330 and elsewhere, regarding the use of altitude versus elevation: Altitude is height relative to the ground. Elevation is height above sea level. Because the ground ranges in elevation depending on where you are in the world, the term elevation is more meaningful for ecological contexts. Please change to elevation throughout the paper.
Ln 333 – change to “ranges that are subject to varying thermal regimes and exhibit substantial..”
Ln 337-338 – What other studies? Please cite the important ones here.
Ln 340 – Our results concur with past findings
Ln 343 – active at the surface
Ln 354-355 – change to “studies on C. horridus that emphasize responses to thermal extremes are needed to elucidate..”
Ln 356 “afforded us to examine” sounds strange. Change to “Including latitude as a model parameter allowed us to examine..”
Ln 357 – insert “that” after “Acknowledging..”
Ln 366 – change to “probability of detection by cameras.”
Ln 375 – rather than say “as reported by studies across the species’ geographic range” I would prefer if you said something like “our top model accurately predicted the average date of emergence for six other populations across the geographic range.”, or something to that effect. It provides more information and is less likely to be misconstrued.
Ln 378 – Please briefly point out whether the studies you used to validate the model used the same techniques. If there were methodological differences you should discuss them. This could certainly be relevant.
Ln 383 – “We also see” sounds strange. You should say something like “Delayed emergence has also been associated with ..”
Ln 384 – Provide the month and day in addition to the Julian day
Ln 385 – spell out West Virginia
Ln 392 – “milder” is ambiguous – it could be interpreted as warmer. Change to read “cooler”
Ln 397 – replace “extirpations” with “those that were extirpated”; also, should “northwestern” read “northeastern”?
Ln 399 – change to read “at the two southernmost sites, in coastal South Carolina..” Note the more meaningful descriptor. County names are not helpful to the average reader.
Ln 401 – insert “that” after “strategy”
Ln 402 – Change to read “The study by Andrews and Waldron (2017) occurred between..”
Ln 403 – delete “to fall”; also, is “lower East Coast states” the best term? The term “southeastern coastal plain” is more meaningful and more precise.
Ln 408 – delete “much”, it is ambiguous and doesn’t add meaning
Ln 409 – replace “represents with “corresponded with”
Ln 411 – replace “but” with “although”
Ln 415-419 – give common names for species discussed here
Ln 420 – change to “..Tennessee based on spikes in..”
Ln 424 – change to read “.., an idea that was partially supported..”
Ln 425 – insert a comma after Jersey County
Ln 435 – insert “that were” before “surgically”
Ln 439 – change to “..may have indicated Snake Fungal Disease..”
Ln 440 – change to read “contusions of unknown origin that may have motivated snakes to emerge and bask.”
Ln 448 – insert semicolon after “conservation”
Ln 451 – change to “elevational”
Ln 454-455 – Change to read “However, predictions about emergence dates at unstudied sites should be treated as hypotheses until verified with empirical data.”
Ln 457 – change to “dates that are useful for”
Ln 458 – change “around” to “near”; “around” could be interpreted to mean activities conducted in a way that stays away from the hibernaculum
Ln 461 – the word “subsequent” can be deleted
Ln 461 – chanve “Consequently” to “Therefore”
Ln 462 – change to read “risk deemed acceptable given a specific”
Ln 463 – change to “latitude in Illinois”
Ln 472-477 – This seems a little premature and contradicts your point that predictions should be treated as hypothesis until wildlife managers have empirical data. I think more validation is needed before wildlife managers start making management decisions based solely on these predictions. Thus, I would revise this section to point out that predictive models, once refined and validated using a larger number of sites across though range, could be used to help wildlife managers more safely schedule risky management activities.
Ln 478 – “methodologies” is just a fancy sounding synonym for “methods”, so please use the simpler of the two
Ln 490 – change to read “population-level risk, but we acknowledge obtaining such information for this enigmatic species would be time- and energy-intensive..”
Ln 498-500 – you may wish to update this accordingly to thank the additional reviewers and editors
Ln 527 - Figure 5 legend – letters correspond to studies listed in Table 6, not Table 5.
Table 3 – the date format used in this table leave room for confusion. Change from the format “6/1” to the format “1-Jun.”
Table 6 – I would change the day of egress from a Julian date to the format 1-Jun.
Table 7 – This table is pretty difficult to digest, and the date format need to be changed to an alpha-numeric Day-Month, which would be hard to fit. Given the space constraints you could simply provide the Julian day. I would consider placing this information into supplementary material.

---

## Round 0.2 · Minor Revisions

Dear Authors,

We are pleased to inform you that both reviewers have acknowledged the thoroughness of your observations and the significant improvements made in the manuscript. Reviewer One has already given their acceptance for the manuscript. Nevertheless, Reviewer Two has brought to our attention the need for some additional citations in the text and the inclusion of certain references from the cited literature. Addressing these minor concerns, as pointed out by Reviewer Two, will likely lead to the final acceptance of your manuscript.

We appreciate your diligence in revising the manuscript and are confident that these minor adjustments will further enhance the quality and completeness of your work.

Warm regards,

Armando Sunny

Reviewer 1 ·

Basic reporting

Summary for all:

This article was already in good shape when I reviewed it earlier, and it is now even better. The authors have done an admirable and thorough job of responding to all points raised by the reviewers. In the few cases where the authors elected not to make changes, they provided good justification for why they elected to not change the text (most of these were basically stylistic decisions). I re-read the manuscript and found it to be an improved version that seems ready for publication.

Experimental design

See above

Validity of the findings

See above

Additional comments

See above

·

Basic reporting

The authors have responded appropriately to the concerns I raised in the initial submission and I am much more satisfied with the paper in it's revised form. However, please note that I found MANY errors when checking to make sure the in-text references matched the reference section. There were so many errors in fact that I stopped checking at line 85. I found 12 places where a source was either missing from the reference section or it had an incorrect year and/or author (e.g. listed as a single author paper in the text but appearing as a multi-author paper in the reference section)...so there are likely to be many others throughout the paper. The entire paper will need to be checked and double checked to fix these kinds of errors.

In-text references missing from the reference section:
Ln 38 Ganz and Pough, 1982
Ln 41 Brown, 1982
Ln 42 Firth, 1998
Ln 48 Gregory, 1988; Macartney and Martin, 1993; Viitanen, 1968
Ln 55 Lutterschmidt, 2006 (perhaps should be et al)
Ln 59 Brown, 1992
Ln 61 DeGregorio et al., 2016
Ln 85 Brown, 1992; MacGowan et al., 2017; Sealy, 2002 (also, there is a comma missing after Sealy)

Problems in the reference section:
Ln 555 should have a hanging indent
Ln 564 I think this should appear after Brown, 199? if it is the same author
Ln 585 Burger, 2019 is listed twice (also on line 579).
Ln 654 Insert a space before "Applied.."
Ln 684 missing volume

Experimental design

The authors have provided enough information in the revised draft to satisfy my questions regarding their modeling approach. I'm satisfied with the changes.

Validity of the findings

More careful language used in the revised manuscript have strengthened the paper.

---

## Round 0.3 · accepted · Accept

Dear Authors,

I am pleased to share excellent news regarding the status of your manuscript. Following a meticulous review process, one of our esteemed reviewers has formally accepted the manuscript. Additionally, the second reviewer noted that implementing the corrections to the cited literature would render the manuscript suitable for acceptance. I am delighted to inform you that these anticipated adjustments have indeed been made.

With immense satisfaction, I convey that your manuscript is now officially accepted for publication in PeerJ. Your dedication to addressing the reviewers' comments and refining your work has certainly paid off.

We extend our sincere appreciation to you for considering PeerJ as the platform for sharing your intriguing manuscript with the scientific community. Your contribution is valued, and we eagerly await the dissemination of your research.

Warm regards,

Armando Sunny